

# Constraining emissions of volatile organic compounds from western US wildfires with WE-CAN and FIREX-AQ airborne observations

Lixu Jin[1], Wade Permar[1], Vanessa Selimovic[1], Damien Ketcherside[1], Robert J. Yokelson[1], Rebecca S. Hornbrook[2], Eric C. Apel[2], I-Ting Ku[3], Jeffrey L. Collett Jr[3], Amy P. Sullivan[3], Daniel A Jaffe[4,5], Jeffrey R. Pierce[3], Alan Fried[6], Matthew M. Coggon[7], Georgios I. Gkatzelis[7, 8, *], Carsten Warneke[7], Emily V. Fischer[3], Lu Hu[1]

1 Department of Chemistry and Biochemistry, University of Montana, Missoula, MT, USA
2 Atmospheric Chemistry Observations and Modeling Laboratory, National Center for Atmospheric Research, Boulder, CO, USA
3 Department of Atmospheric Science, Colorado State University, Fort Collins, CO, USA
4 School of Science, Technology, Engineering and Mathematics, University of Washington, Bothell, WA, USA
5 Department of Atmospheric Sciences, University of Washington, Seattle, WA, USA
6 Institute of Arctic and Alpine Research, University of Colorado, Boulder, CO, USA
7 Chemical Sciences Laboratory, National Oceanic and Atmospheric Administration, Boulder, CO, USA
8 Cooperative Institute for Research in Environmental Sciences, University of Colorado, Boulder, CO, USA
* Now at: Institute of Energy and Climate Research, IEK-8: Troposphere, Forschungszentrum Jülich GmbH, Jülich, Germany

Correspondence: Lixu Jin (lixu.jin@umconnect.umt.edu); Lu Hu (lu.hu@mso.umt.edu)

**Abstract.** The impact of biomass burning (BB) on the atmospheric burden of volatile organic compounds (VOCs) is highly uncertain. Here we apply the GEOS-Chem chemical transport model (CTM) to constrain BB emissions in the western US at ~25 km resolution. Across three BB emission inventories widely used in CTMs, the total of 14 modeled BB VOC emissions in the western US agree with each other within 30–40 %. However, emissions for individual VOC differ by up to a factor of 5 (*i.e.,* lumped $\geq C_4$ alkanes), driven by the regionally averaged emission ratios (ERs) among inventories. We further evaluate GEOS-Chem simulations with aircraft observations made during WE-CAN (Western Wildfire Experiment for Cloud Chemistry, Aerosol Absorption, and Nitrogen) and FIREX-AQ (Fire Influence on Regional to Global Environments and Air Quality) field campaigns. Despite being driven by different global BB inventories or applying various injection height assumptions, GEOS-Chem simulations underpredict observed vertical profiles by a factor of 3–7. The model shows small-to-no bias for most species in low/no smoke conditions. We thus attribute the negative model biases mostly to underestimated BB emissions in these inventories. Tripling BB emissions in the model reproduces observed vertical profiles for primary compounds, *i.e.,* CO, propane, benzene, and toluene. However, it shows no-to-less significant improvements for oxygenated VOCs, particularly formaldehyde, formic acid, acetic acid, and lumped $\geq C_3$ aldehydes, suggesting the model is missing secondary sources of these compounds in BB-impacted environments. The underestimation of primary BB emissions in inventories is likely attributable to underpredicted amounts of effective dry matter burned, rather than errors in fire detection, injection height, or ERs. We cannot rule out potential sub-grid uncertainties (*i.e.,* not being able to fully resolve fire plumes) in the nested GEOS-Chem which could explain the model negative bias partially, though the back-of-the-envelope calculation





and evaluation using longer-term ground measurements help increase the argument of the dry matter burned underestimation. The ERs of the 14 BB VOCs implemented in GEOS-Chem account for about half of the total 161 measured VOCs (~75 versus 150 ppb ppm-1). This reveals a significant amount of missing reactive organic carbon in widely-used BB emission inventories. Considering both uncertainties in effective dry matter burned and unmodeled VOCs, we infer that BB contributed up to 10 % in 2019 and 45 % in 2018 (240 and 2040 GgC) of the total VOC primary emission flux in the western US during these two

fire seasons, compared to only 1–10 % in the standard GEOS-Chem.

## 1 Introduction

Biomass burning (BB), including wild and prescribed fires, is estimated to be the largest primary source of fine particulate matter (PM) and the second largest source of volatile organic compounds (VOCs) globally (Yokelson et al., 2008), impacting air quality, public health, and climate. In fire-prone areas such as the western United States (US), the relative importance of

BB emissions as a source of air pollution has been growing due to increased wildfire activity (Westerling, 2016; Higuera et al., 2021) and decreased anthropogenic emissions (Warneke et al., 2012; Simon et al., 2015). Wildfires have been suggested to account for up to half of the overall $PM_{2.5}$ burden since 2012 and contribute to its increasing trend in the last three decades in the western US (McClure and Jaffe, 2018; O'Dell et al., 2019; Burke et al., 2021). Wildfire impacts on VOC burdens are highly uncertain, in part due to the limited observational constraints on BB VOC emissions. Here we apply comprehensive

VOC observations from two recent aircraft campaigns targeting fires, along with the GEOS-Chem chemical transport model (CTM), to examine our understanding of BB emissions in the western US.

Current CTMs often poorly simulate the impact of wildfire smoke partly because of an incomplete description of the amount and speciation of VOC emissions, along with poor representation of their spatial, temporal, and vertical distributions (Alvarado

and Prinn, 2009; Jaffe and Wigder, 2012; Jaffe et al., 2018; Baker et al., 2016, 2018; Wolfe et al., 2022). BB emission estimates are typically derived from the product of a compound-and-biome-specific emission factor (EF, expressed as mass of species in g per dry biomass burned in kg) and an effective amount of dry matter burned (effective DM burned, kg). Both EF and DM burned are subject to large uncertainties. EFs are either measured in laboratory burning experiments that attempt to simulate real-world fires, or quantified from near-field measurements on the ground or air that may be influenced by atmospheric aging

processes before sampling (*e.g.,* Burling et al., 2010; Warneke et al., 2010; Wooster et al., 2011; Permar et al., 2021; Majluf et al., 2022). Recent efforts to reconcile the difference between laboratory and field measurements support the need to adjust lab EFs to the typical field combustion efficiency (Permar et al., 2021; Selimovic et al., 2018). However, the burn conditions throughout the course of a fire are currently not considered in inventories. In addition, commonly used global BB emission inventories often consider only 3–6 biome groups (Andreae and Merlet, 2001; Wiedinmyer et al., 2011; Akagi et al., 2011;

Randerson et al., 2012; Kaiser et al., 2012; Koster et al., 2015; Andreae, 2019). For example, the Quick Fire Emissions Database version 2.4 (QFED2.4) inventory has three biome groups to represent all global biomass: tropical forest, extratropical





forest, and savanna/grass (Koster et al., 2015). In the Global Fire Emissions Database version 4 with small fires (GFED4s), the extratropical forest biome is subdivided into the boreal forest and temperate forest, and additional two biomes for peatland and agriculture/waste burning are considered, thus a total of six (van der Werf et al., 2017). Differences (and errors) in

vegetation classifications among inventories can also lead to diverse assigned EFs, even though those EFs may come from the same experimental studies, thus resulting in different emission estimates.

DM burned in emission inventories is estimated by two different satellite remote sensing approaches. The 'bottom-up' method estimates DM burned from the product of fire burn areas (BA) and fuel consumption (*i.e.*, loading, type, timing, and rate).

Global BB emission inventories using this method include GFED4s (van der Werf et al., 2017) and the Fire INventory from NCAR version 1.5 (FINNv1.5; Wiedinmyer et al., 2011). The 'top-down' approach uses the fire radiative power (FRP, radiant energy released per time by burning fuel) and its empirical relationship with biomass burned. Some widely used BB emission inventories using this approach include QFED2.4 (Koster et al., 2015) and the Global Fire Assimilation System version 1.2 (GFASv1.2; Kaiser et al., 2012). Both top-down and bottom-up inventories share common sources of uncertainties, such as

missing fire detections and/or FRP observations used to initialize DM burned estimates. Additionally, those fire products are mostly from polar orbiting satellites with a low temporal coverage (*i.e.*, once or twice daily at a fixed local time) and can be obscured by clouds and smoke, resulting in assumptions often have to be made to fill both temporally and spatial gaps in the observations (Wang et al., 2018; Wiggins et al., 2020; Stockwell et al., 2022). Current operational BB emission inventories can produce monthly CO and aerosol fluxes that vary by a factor of 5 or even 20 for a specific region (Al-Saadi et al., 2008;

Zhang et al., 2014; Koster et al., 2015). These differences in global total emissions averaged over longer periods are smaller, but still on the order of a factor of 2–4 (Stroppiana et al., 2010; Granier et al., 2011; Carter et al., 2020; Liu et al., 2020; Pan et al., 2020). The discrepancy could be even larger in VOC emission estimates due to different speciation among inventories (*i.e.*, GFED4s has 21 VOCs while QFED2.4 has 9 VOCs). Different input data used to drive BB emissions, such as EFs, fire detections, fire burned area, and the amount of biomass burned are all thought to contribute to the divergent estimates among

emission inventories. Recently, Carter et al. (2020) suggested that, at least for aerosol, the BB emission uncertainties are mostly from DM burned at both regional and global scales; and that differences in EFs across inventories are smaller than differences in DM burned. These errors in estimating DM burned will also affect VOC emission estimates thus their uncertainty is thought to be at least at a similar order as that of aerosol and CO.

When compared to observations, previous model evaluation studies (again mostly focusing on CO and aerosol) often point to a general underestimation of BB emissions in the commonly used inventories, and a factor of 2 as the global BB emission uncertainty (Kopacz et al., 2010; Wang et al., 2018; Carter et al., 2020; Pan et al., 2020; Bela et al., 2022). For example, various degrees of negative model bias are found in aerosol optical depth and CO near BB source regions when compared to corresponding satellite and ground observations, though the FRP based BB inventories often provide higher (more accurate)

emissions than the bottom-up estimates (Yurganov et al., 2011; Petrenko et al., 2012, 2017; Zhang et al., 2014; Reddington et



al., 2016; Pan et al., 2020; Liu et al., 2020; Bela et al., 2022). For the western US, Pfister et al. (2011) suggested that an early version of FINN (version 1) underestimated BB emissions by a factor of 4 over California as revealed by constraints from aircraft and satellite measurements. More recently, Carter et al. (2021) found that the GEOS-Chem model driven by GFED4s is biased low for CO but captures the carbonaceous BB aerosol when compared to the recent WE-CAN airborne observations

(Western Wildfire Experiment for Cloud Chemistry, Aerosol Absorption, and Nitrogen). Another recent study by Bela et al. (2022) shows that, though bracketing the observed BB CO fluxes, emission estimates from seven existing inventories span a factor of 83 in their daily mean for a case study of a western US wildfire. Even with observational constraints on certain input parameters (*e.g.,* for relating FRP to the amount of biomass burned or emissions released), their uncertainty range is still a factor of ~2 compared to the direct CO flux measurements in fire plumes (Bela et al., 2022). A similar case study also suggested

a wide spread of the hourly emission estimates (spanning a factor of > 33) from nine satellite-based inventories in the FIREX-AQ airborne observations (Fire Influence on Regional to Global Environments and Air Quality) (Stockwell et al., 2022).

Here we aim to improve current understanding of VOC emissions from wildfires in the western US. Leveraging the comprehensive VOC observations from the WE-CAN airborne campaign, we evaluate a $0.25° \times 0.3125°$ nested version of the

GEOS-Chem CTM driven by three commonly used global BB emissions inventories (Sect. 4). We assess the potential reasons for model and observation discrepancies including the fire detections, emission ratios, and plume injection heights used in the emission inventory/CTM (Sect. 5 and 6). We further apply independent measurements from ground sites and the FIREX-AQ airborne campaign to test the regional representativeness and interannual variability of our findings (Sect. 7).

## 2 Methods

**2.1 WE-CAN aircraft campaign**

The WE-CAN airborne campaign systematically characterized emissions and chemical evolution of western US wildfire smoke with the NSF/NCAR C-130 research aircraft. The campaign was mainly based in Boise, ID in July-September 2018 and sampled 27 fire plumes in the near field (some fires measured multiple times on different days), and various cases of regional and aged smoke (Lindaas et al., 2021; Permar et al., 2021). Table S1 summarizes the sampling time, fire location, and

acres burned for specific fires sampled during WE-CAN.

Four sets of complementary VOC measurements were utilized to constrain BB emissions, including a proton-transfer reaction time-of-flight mass spectrometer (PTR-ToF-MS, or PTR), trace organic gas analyzer (TOGA), advanced whole air sampler (AWAS), and iodide (I⁻) adduct high-resolution time-of-flight chemical-ionization mass spectrometer (I-CIMS). The four

instruments have different strengths and weaknesses in terms of analytical and separation powers, uncertainty, and measurement frequencies (Apel et al., 2010; Andrews et al., 2016; Palm et al., 2019; Permar et al., 2021).





We primarily focus on 14 VOCs or lumped VOC groups that are represented in the standard GEOS-Chem version 12.5.0 with observations assigned to the model speciation (Tables 1 and S2). Among them, three VOCs were mostly measured by discrete sampling with AWAS and in emission transects (nearest downwind with < 2 hours aging). Thus we limit their model evaluation

to emission ratios. These include ethane, lumped alkanes with four or more carbon atoms (or lumped ≥ $C_4$ alkanes) and lumped ≥ $C_3$ alkenes. The other 11 VOC measurements used higher frequency instruments, allowing for more comprehensive model evaluations along the C-130 (and DC-8) flight tracks. For these, we follow the data reduction described in Permar et al. (2021) mainly using PTR data with interferences corrected using co-deployed TOGA measurements and laboratory observations (Koss et al., 2018).


Figure S1 compares key VOCs measured by higher frequency instruments in the entire WE-CAN (and FIREX-AQ) datasets. We find that PTR agrees with I-CIMS within ± 20–40 % for formic acid. PTR agrees with TOGA measurements within ~20 % for formaldehyde, acetaldehyde, acetone, MEK, benzene, and toluene, with high correlation between each instrument ($r$ = 0.93–0.99; and similar agreements are found in the FIREX-AQ dataset). PTR measured xylenes is ~60% higher than in TOGA

during WE-CAN (and lower by 20% in FIREX-AQ), but again they are highly correlated to each other. The difference in xylenes measurements is possibly due to unknown fragmentation and/or under-characterized instrument sensitivity from likely varying isomer fractions in smoke plumes in PTR. We will discuss such measurement uncertainty and how it may affect the conclusions of this work (Sect. 4).

The emission ratios relative to CO in WE-CAN emission transects identified in Permar et al. (2021) are used to evaluate this

key input in emission inventories. CO was measured at 1Hz with 1 ppb accuracy with a Picarro G2401-m WS-CRDS analyzer during WE-CAN. All observations were taken from the WE-CAN 1-minute merge data unless otherwise noted (version 4; https://www-air.larc.nasa.gov/cgi-bin/ArcView/firexaq?MERGE=1).

**2.2 FIREX-AQ aircraft campaign and ground sites data**

Two additional datasets are used to examine the broader representativeness and the year-to-year variability of our findings

from WE-CAN. We use ground-level CO mixing ratios from nine western US sites measured during WE-CAN 2018 to assess the model prediction of regional BB emissions at the surface over the fire season. These include a mountaintop site at Mt. Bachelor Observatory, OR, a long-term ground station in Missoula, MT, and seven available EPA monitoring stations across the western states (Table S3 and Fig. 1; Laing et al., 2017; Selimovic et al., 2020; https://www.epa.gov/aqs).

We also repeat the WE-CAN analyses using FIREX-AQ DC-8 aircraft observations that took place in July-September 2019. FIREX-AQ was a joint field campaign led by NOAA and NASA that investigated the chemistry and transport of smoke from both wildland and agricultural fires in 2019. Here we focus on the western US portion of FIREX-AQ, which represents 64 % of the entire campaign data (Fig. 1 and Table S4). The DC-8 in FIREX-AQ systemically sampled 18 wildfires in the western US and here we use the 1-minute merge data unless otherwise noted (version R1; https://www-air.larc.nasa.gov/cgi-



bin/ArcView/firexaq). The wildfire emission sizes during FIREX-AQ were less than during WE-CAN as reflected by the GFAS total VOC emissions (20 GgC versus 190 GgC in the western US on campaign-specific days) and the distribution of measured acetonitrile abundance in both campaigns (Fig. S2). Together with the surface CO measurements, they provide independent evidence to test if the model emission biases found from WE-CAN in 2018 are representative across the western US and in different years.

**2. 3 GEOS-Chem chemical transport model**

We employ GEOS-Chem nested grid simulations (version 12.5.0; Bey et al., 2001; www.geos-chem.org; http://doi.org/10.5281/zenodo.3403111) to interpret the recent airborne observations and ground measurements in terms of new constraints on western US VOC emissions from wildfires. GEOS-Chem is driven by assimilated meteorology from the NASA Goddard Earth Observing System (GEOS). Here we use GEOS-FP meteorological inputs to drive GEOS-Chem nested

grid simulations over North America for the WE-CAN (24th July–14th September 2018) and FIREX-AQ periods (22nd July–5th September 2019). The nested domain covers 10°–70°N and 140°–60°W, with 0.25° × 0.3125° (~25 km × 30 km; latitude × longitude) horizontal resolution and 47 vertical layers extending up to 0.01 hPa (Wang et al., 2004; Kim et al., 2015). The model boundary conditions are obtained from the global simulation at 4° × 5° resolution every 3 hours. The transport/convection and emission/chemistry time steps of the nested simulation are 5 min and 10 min, respectively. We carry

out a full year spin-up simulation at 4° × 5° resolution followed up by another one-week spin-up at the nested resolution prior to the time periods of interest, to minimize effects from initial conditions.

The GEOS-Chem chemical mechanism includes detailed $HO_x$-$NO_x$-VOC-ozone-halogen-aerosol chemistry with fully coupled troposphere and stratosphere (Park, 2004; Mao et al., 2010; Eastham et al., 2014; Schmidt et al., 2016). Dry deposition uses a

standard resistance-in-series model (Wesley, 1989). Wet deposition includes scavenging of soluble tracers in convective updrafts, as well as rainout and washout of soluble tracers (Liu et al., 2001). Emissions are computed using the HEMCO module described by Keller et al. (2014). These include biogenic VOC emissions from the MEGANv2.1 (Guenther et al., 2012) as implemented by (Hu et al., 2015b), and anthropogenic emissions from the CEDS global emission inventory overwritten with the EPA's national emission inventory 2011 (NEI 2011) for the US (Hoesly et al., 2018). Below we describe

aspects of the model configurations that are most relevant to this work.

We carried out several simulations driven with four different global BB emission inventories. An initial result suggests that the FINNv1.5 emission inventory predicted only 4–8 % of western US BB VOCs or CO emissions as those from the other three inventories, even though their total global emission estimates agree within 40 %. This is likely due to fuel characterization

errors for this region in FINNv1.5, thus we focus on simulations with GFED4s, GFASv1.2, and QFED2.4 for the analysis in this work. A recent study found that these three inventories strongly correlate with aircraft-derived hourly total carbon



emissions during FIREX-AQ, but generally underpredict BB and cannot capture the observed fire-to-fire variability (Stockwell et al., 2022).

For simplification, we denote these three BB inventories as GFED4, GFAS, and QFED in the following discussion. We also note that the BB emission inventories, in the standard GEOS-Chem, may not contain a complete list of VOCs in the model. We thus implement their BB emissions in the base simulation (GEOS-Chem + GFAS; Table 2) by scaling the CO BB flux with WE-CAN field measured ERs from Permar et al. (2021). These species include MEK, formic acid, acetic acid, and lumped $\geq C_3$ aldehydes in GEOS-Chem + GFAS (Table 1).


The standard GEOS-Chem version also implements different emission injection height schemes for each BB inventory, providing an opportunity to examine the impact of various plume height assumptions on the vertical distribution of trace gases. Specifically, GFED4 (and FINNv1.5) emissions, as incorporated in GEOS-Chem, are prescribed in the model surface layer, and mainly rely on diffusion and convection (which depends on atmospheric turbulence and stability), for mixing before the

chemistry operator. QFED prescribes 65 % of BB emissions by mass evenly from the surface to the top of the planetary boundary layer (PBL), and the remaining 35 % are evenly distributed between the PBL height and 5500 m. This approach was based on the distribution pattern of aerosol smoke plume heights from 5-year Multi-angle Imaging Spectro Radiometer (MISR) observations and was suggested to improve PAN simulations at high latitudes (Val Martin et al., 2010; Fischer et al., 2014).

GFAS, as implemented in the standard version of GEOS-Chem, releases emissions evenly from the model surface to the mean altitude of maximum injection ('mami'). GFAS also provides estimates of the top and the bottom of the plume at its native resolution (0.1° × 0.1° and daily). All three products are derived from the Moderate Resolution Imaging Spectroradiometer (MODIS) FRP product and a plume rise model (PRM) at GFAS native pixels (Latham, 1994; Freitas et al., 2007). The PRM model uses atmospheric profiles of meteorological parameters and fire information from European Centre for Medium-Range

Forecasts (ECMWF) and MODIS observations to derive a full smoke detrainment profile and further to be translated into injection height information (Rémy et al., 2017). The BB-free region is regarded as plume heights of zero in the model. Thus, the plume heights would be artificially reduced when averaging to the coarser-than-native-resolution (*i.e.,* 0.25° × 0.3125° here). To account for this grid-dependent issue, we calculate emission-flux-weighted averages for those GFAS plume height products at corresponding GEOS-Chem resolution.


For GFAS and QFED with temporal resolution that vary daily, the standard GEOS-Chem prescribes a climatologically diurnal distribution profile that emits the majority (~85 %) of the daily BB emissions in the afternoon (local time) (Western Regional Air Partnership, 2005). For GFED4 with monthly temporal resolution, the model distributes the daily fraction using MODIS active fire products and climatological mean diurnal cycles (Mu et al., 2011). These temporal patterns are in general consistent

with observations in the western US as wildfires tend to be most active in the afternoon. In this work, we do not attempt to constrain the diurnal distribution of BB emissions as that would require continuous observations in the near field or from space. We note that a recent study found that varying diurnal distribution using FRP observed from a geostationary satellite (so that diurnal cycles of BB emissions vary from grid to grid and from day-to-day) shows little improvement compared to the climatological approach at least in the western US (Tang et al., 2022).


Table 2 summarizes all the simulation experiments used in this study. These include three default simulations driven by the different emission inventories which all have different plume height schemes in the standard GEOS-Chem ('Inventory experiments'). In addition, we employ five different plume injection schemes in combination with the GFAS to test assumptions regarding BB emission vertical distribution ('Injection experiments'). Further, we carry out one simulation with

BB emissions turned off ('noBB') and another simulation with 3 times the default GFAS BB emissions ('3 × GFAS') as additional sensitivity tests to examine the BB impact in the western US. All the simulations were performed for the summer of 2018 and 2019, covering both the WE-CAN and FIREX-AQ campaign periods.

To directly compare the model to the aircraft observations, the model outputs are sampled along the C-130 and DC-8 flight

tracks (same location and altitude) and at the time of the flights in every minute. Then both observations and model results are averaged to the center of the model horizontal grid boxes and to transport resolution (0.25° × 0.3125°; 5 min). It is a general concern that Eulerian models are not able to resolve sub-grid features partly resulting from point source emissions needed to dilute instantly to relatively coarse model grid sizes, particularly when compared to observations from aircraft campaigns targeting fresh fire smoke plumes. Thus our model evaluation doesn't focus on individual fire cases, but rather on the campaign

average conditions, no/low smoke environments, and trace:trace ratios. In addition, we apply the ground-based measurements over a longer term as an additional test as they should be less sensitive to any potential model biases due to not properly accounting for sub-grid features in simulating BB emissions. For this purpose, daily averaged CO observations from nine western US ground sites throughout the summer 2018 are used to evaluate model outputs as an extra representativeness validation. For this, either model outputs at the surface layer or the corresponding elevation of observations (*i.e.,* Mt. Bachelor

Observatory) are used.

## 3 Current knowledge of VOC emissions in the western US

Figure 1 shows the VOC primary emissions over the western US in the base simulation during the 2018 fire season (June-September, or JJAS). These four months typically account for 70–90 % of the annual acreage burned in this region (Jaffe et al., 2008). In GFAS, the JJAS contributes 85 % of the 2018 annual BB emissions in the western US. According to emission

inventories chosen in the GEOS-Chem, biogenic emissions are thought to be the dominant VOC source in summer in this region (2200 GgC or ~75 % of the total VOC emissions), followed by anthropogenic emissions (405 GgC or ~15 %), and BB



emissions. In 2018 JJAS, the total BB emissions from 14 VOCs in the model range from 220 to 340 GgC (or 10 % of the total). As we will show later, our model:observation comparisons suggest that the role of BB is significantly underestimated in the current CTMs.


In the 2019 fire season, the BB VOC emissions (40–75 GgC or 1–3 % of total primary VOC emissions) are 10–30 % of that in 2018 for this region, and about 40–50 % of the 2019 annual BB emissions depending on which BB inventory is used. This shows the WE-CAN and FIREX-AQ aircraft campaigns sampled two distinct fire seasons which may reflect upper and lower bounds of wildfire activity in this region (https://www.nifc.gov/fire-information/statistics/wildfires). Despite the large

interannual variability of wildfire emissions, the western US accounts for ~90 % of BB VOC emissions in the contiguous United States (CONUS) in 2018 and ~60 % in 2019 according to GFAS, confirming a significant fire influence exists in the western US, which could also affect the rest of CONUS downwind (O'Dell et al., 2021).

The total BB VOC emission estimates in the western US differ by 20–40 % across the three global inventories examined for

the 2018 fire season. All emission inventories show similar spatial distributions as they all use MODIS satellite products such as active fire, burned areas, or FRP as inputs. However, larger differences between inventories occur for emission estimates for individual fires on specific days (more than a factor of 20), also shown in Bela et al. (2022) and Stockwell et al. (2022). These differences likely reflect the various assumptions or adjustments made for fire persistence, small fires, or fires obscured by clouds and haze in the inventories (Liu et al., 2020).


All three global BB inventories suggest aldehydes, alkanes, and alkenes are the most abundantly emitted VOCs from western US wildfires, largely consistent with recent field measurements (Permar et al., 2021). However, emission estimates for individual VOCs disagree by a factor of 1–5 in the western US fire season (Figs. 2 and S3). Emission estimates for xylenes show the largest difference (5 GgC in GFED4 versus 1 GgC in GFAS), while propane emissions agree within ± 20 % across

the 3 inventories (8–10 GgC).

A recent global study comparing BB aerosol emissions from inventories suggests that the effective DM burned is the biggest contribution to divergent emission estimates across inventories (Carter et al., 2020). In contrast, we find that the regionally averaged ERs dominate disagreement in emission estimates for most VOCs across the three inventories (Fig. 2). These ERs

are regionally averaged from each inventory, thus are functions of both assigned ERs for specific biome and vegetation classifications. They are calculated from the regression of daily mean VOC and CO BB emission fluxes at each grid cell for the region from inventories. We infer that, at least for the western US, these inventories agree on the amount of effective DM burned within 40 % (47–67 Tg in 2018, as calculated by dividing VOC and CO emission estimates with corresponding regionally averaged EF). Using the National Interagency Fire Center burned area report (~2,420,000 ha for 2018 in the west),

the back-of-the-envelope calculation suggests that these global inventories' effective DM burned per area is 19–28 Mg ha$^{-1}$.



These values of biomass burned per area are lower than the USFS model FOFEM estimates for western US wildfires (~110 Mg ha⁻¹; Reinhardt et al., 1997), and field measurements of western mixed conifer forest wildfires (~32–44 Mg ha⁻¹; Campbell et al., 2007; Hyde et al., 2015), indicating the underestimation of DM burned in these three BB inventories.

## 4 Model evaluation with WE-CAN aircraft observations

Figures 3 and 4 show the vertical distribution of CO and VOCs sampled by the C-130 during WE-CAN, and the comparisons to various simulations. The observed abundance of all species is elevated by 50–300 % within the planetary boundary layer (> 850 hPa), indicating influences from anthropogenic, biogenic, and/or BB emissions near the surface during take-off and landing time. The higher abundance in the middle troposphere (750–500 hPa) than typical background conditions (*i.e.,* < 500 hPa) are mostly due to BB, as the C-130 targeted sampling wildfire smoke in both near-field, and aged smoke whenever
feasible while in transit during WE-CAN.

Simulations driven by different BB emission inventories show remarkably similar abundance (mostly within ± 10 %, except for surface toluene and CO within ± 30–40 %). All the inventories capture the enhancement patterns observed by the C-130, both elevated altitudes and timing with high correlations with observations ($r = 0.7$–$1.0$ in 5-min averaged data). The sensitivity
run with no BB emissions (noBB) indicates that wildfire is a significant source for CO and primary VOCs including propane, benzene and toluene during WE-CAN (enhanced by 2–3 times compared to noBB), but a lesser source for OVOCs especially for formaldehyde (Figs. 4 and S4). The model driven by GFAS (GEOS-Chem + GFAS) tends to simulate slightly higher and better VOC than GFED4 and QFED, possibly reflecting that GFAS has more accurate ERs as discussed later in Sect. 6.

All 3 inventory experiments significantly underestimate observed CO and VOCs except for MEK. In the middle to lower troposphere (> 500 hPa), simulations reproduce 40–70 % of the observed abundance of CO, benzene, toluene, and acetone, and 30–40 % of the observed propane, formaldehyde, acetaldehyde, and lumped ≥ C$_3$ aldehydes, but only 0–10 % of the observed organic acids. The model suggests mixed performance for xyelnes, *i.e.,* a high bias of 0–100 % in the lower troposphere and a low bias of 50–100 % in middle troposphere. In a relatively clean environment (< 500 hPa), the simulations
show relatively small negative biases for all compounds and tend to match observations in generally clean or well-mixed environments during WE-CAN.

Unlike other VOCs and CO, MEK is systematically overestimated by 50–300 % throughout the middle to lower troposphere in all simulations including noBB, but being reproduced in a relatively clean environment (< 500 hPa). Similar positive model
bias has been reported in a recent study comparing GEOS-Chem to a comprehensive suite of airborne datasets over North America (Chen et al., 2019). This is likely due to the overestimation of MEK or its precursors in the EPA NEI and/or MEGAN




inventories (Yáñez-Serrano et al., 2016), as such large high model bias exists even when the BB influence is removed (Fig. 5). Thus, further evaluation is needed for the source MEK and its precursors in anthropogenic and biogenic emission inventories.

We further refine the analysis in low/no smoke conditions by filtering out data when either the observed acetonitrile (CH₃CN) mixing ratio, a known BB tracer, is > 159 ppt (25th quantile of CH₃CN) or the enhancement ratio of CH₃CN relative to CO is > 2.01 ppb ppm⁻¹ (Huangfu et al., 2021). The vertical profiles after applying this filter are shown in Figs. 5 and S5 and represent about one third of the sampling time during WE-CAN, allowing us to examine the non-BB related processes/emissions. Compared to the full campaign data, the observations of CO and all VOCs in low/no smoke conditions are lower by a factor

of 2 or more, confirming the important influence from BB in the western US during WE-CAN. The simulations capture the observed CO, benzene, and toluene in this clean environment, but still underestimate the rest of the VOCs (especially OVOCs) by 10–90 % except MEK. The model low bias for formaldehyde in the free troposphere can be partly due to underestimated oxidation of CH₄ or other precursors (Zhao et al., 2022). The model negative bias for acetaldehyde, formic acid, and acetic acid in the PBL may be related to missing or underestimated precursors from biogenic emissions (Millet et al., 2010, 2015;

Paulot et al., 2011). The negative bias for acetone in the middle-upper troposphere may reflect a poorly constrained global background from ocean sources in GEOS-Chem (Wang et al., 2020). Nevertheless, the negative model bias in the low/no smoke conditions sampled during WE-CAN (Fig. 5) is much smaller than the BB-influenced environment. Thus, underestimation in the low/no smoke conditions does not explain model underestimation across compounds in the full campaign dataset (Fig. 4).


We calculate the average model biases that are due to BB processes for each species using the enhancements between the full campaign dataset and the low/no smoke conditions. Given the calculation of primary trace gases (CO, propane, benzene, and toluene), we conclude that the model potentially underestimates BB emissions or related processes by a factor of 3–7 in the GFAS while the bias can slightly vary in the GFED4 or QFED. Thus, we further carry out a sensitivity run by tripling the

GFAS emissions in the model (GEOS-Chem + 3 × GFAS) as a test of the BB impact in the western US. Figures 3 and S4 show that tripling BB primary emissions results in evident improvements and reproduces the observed levels for CO and most primary VOCs (propane, benzene, and toluene). The improvement for xylenes is moderate, due to other model errors in the averaged OH reaction rate constant and ER (Sect. 6).

The GEOS-Chem + 3 × GFAS has elevated simulated abundance for OVOCs to various degrees compared to the base run. For acetaldehyde and acetone, we find that the 3 × GFAS brings the model close to the measurement uncertainty. For formaldehyde, formic acid, acetic acid, and lumped ≥ C₃ aldehydes, tripling the primary BB emission of these species (and their precursors that are included in the model) does not significantly improve the model:observation discrepancy (the difference is within 5 %). Since 3 × GFAS mostly corrects the model error in primary BB emissions, this underestimation

suggests there are likely large secondary sources of these compounds in BB plumes that are missing in the current model.



Eulerian models are known to have trouble preserving sub grid features such as concentrated fire plumes over time due to rapid dissipation by numerical diffusion (Eastham and Jacob, 2017; Rastigejev et al., 2010). Campaigns targeted plumes can get particularly intense, thus deviating from the climatologically diurnal distribution of BB emission used in the model, and

resulting in model low bias when compared to aircraft measurements. In addition, any wind direction or plume height errors in the model would result in the model's aircraft diagnostics missing the fire plume when the real aircraft sampled it, contributing to some amount of a low bias. Finally, if the plumes are narrower than ~25 km (and the aircraft transect lengths are also narrower than ~25 km), then the plume will dilute in the model grid box more than the plane observed (even when including the transect portions outside of the plumes), also contributing to a model low bias. In addition to the BB emissions,

those above factors due to fire sub-grid features may all have contributed to the model low bias in the aircraft analysis but it is difficult to fully tease them out if at all possible. We thus consider the model bias revealed here as the upper limit of BB emission underestimation in the global inventories (Sects. 3 and 8).

## 5 Model uncertainties in fire detection and emission injection heights

To explore causes for the underestimation of BB emission for these three emission inventories, we first determine if the

inventories have detected the 27 individual fire plumes sampled in WE-CAN. A fire is considered to be detected if the inventory registers any CO emissions in the model surface grid box at its location when the C-130 arrived. Table S1 shows that all the BB inventories (including FINNv1.5) capture all the sampled fires. BB emission inventories typically rely on space-based observations of burned area or FRP (*i.e.,* MODIS-Terra and Aqua fire products) for fire detections. For example, MCD64A1 burned area products are applied in GFED4 and MOD14/MYD14 FRP products are used in GFAS and QFED. During WE-

CAN, wildfires were mostly sampled in the late afternoon when fires were the most active. The fires sampled by the C-130 tended to have developed well-defined plumes that were visible from geostationary GOES-16 or 17 GeoColor images in the morning of the same day when flight planning was finalized. Our finding suggests that the fire detection products from low-orbital satellites commonly used in global BB emission inventories are efficient at detecting large fires in the western US that tend to burn for several days if not weeks or months.


We further examine the impact of the assumed injection altitude of BB emissions by conducting sensitivity tests using five different BB injection height schemes (Table 2; Sect. 2.3). Figure S6 shows almost identical model vertical profiles in the five plume injection experiments, particularly in the free troposphere. In the PBL, releasing BB emission at the surface tends to result in the highest surface mixing ratios among the experiments, but the differences across simulations are within ± 10 %

except for benzene and toluene (about ± 40 % near surface). The model does not appear to be highly sensitive to assumptions regarding BB injection heights in the western US at ~25 km resolution. This insensitivity is likely because the trace gas emissions from large wildfires are efficiently lifted into the free troposphere by efficient vertical mixing in the summer (Chen

et al., 2009; Jian and Fu, 2014). However, the choice of plume injection heights can still be important for secondary productions and downwind areas (Tang et al., 2022). For example, daily mean ozone concentrations vary by up to 14 % or 4 ppb at the

surface in our injection experiments. Thus, the impact of various BB emission injection schemes on surface air quality needs further investigation, especially for populated downwind regions.

## 6 Model uncertainties in emissions ratios

Emission ratio (ER; often interchangeable with emission factor or EF) can be a source of uncertainty in BB emissions estimates if they are poorly characterized or unmeasured (*e.g.,* Akagi et al., 2011; Urbanski et al., 2011). We calculate ERs from the

slope of the reduced major axis regression of VOCs and CO measured (and simulated) in emission samples. The 31 fresh BB emission transects identified in WE-CAN and the plume samples with physical age < 1 h in FIREX-AQ are used to calculate ERs. We note the observed ERs derived here using the 5-min averaged data tend to agree with what Permar et al. (2021) reported within 20 %, despite Permar et al. (2021) calculating ERs from the 1 sec observation and using the integration approach. Also, calculating observed and simulated ERs in a consistent way and according to the temporal and spatial

resolution of the model can provide a valuable constraint on the overall model processes in terms of BB emission locations, timing, transport, and chemistry in fire influenced environments.

Figure 6 illustrates this approach with scatterplots of a subset of observed VOCs and CO in emission transects, and their comparison to the simulated relationship in GEOS-Chem + GFAS. The model shows the strong correlations between VOCs

and CO ($r$ = 0.7–1.0), suggesting GFAS captures the regional BB locations and timing sampled by the C-130 (Sect. 5). We find GFAS ERs agree with observed ERs within 30% or better for formaldehyde, acetaldehyde, benzene, toluene, and lumped $\geq C_4$ alkanes. GFAS is either too high or too low by 50–70 % for ethane, propane, and acetone. Overall, GEOS-Chem + GFAS tends to produce higher and more accurate ERs than the other two inventories (Figs. 7 and S7). Some notably large errors in simulated ERs ($\geq$ a factor of 2) include acetaldehyde in QFED, and acetone, MEK, benzene, and toluene in GFED4.


The modeled abundance and ERs of xylenes and lumped $\geq C_3$ alkenes are significantly underestimated across all inventory experiments. These two lumped VOC groups are highly reactive, with lifetimes of ~1 hour (assuming an average in-plume OH concentration of $1 \times 10^7$ molecule cm$^{-3}$ and an OH reaction rate constant $k_{OH}$ of 23.1–25.0 $\times$ 10$^{-12}$ cm$^3$ molecule$^{-1}$ s$^{-1}$). Errors in their loss via OH reactions due to incorrect OH concentration or $k_{OH}$ could distort their simulated abundance and ERs. Model

bias in OH concentration would affect all primary species in the same direction, and reactive VOCs would be particularly sensitive to such error. Thus, we use aromatic hydrocarbon:hydrocarbon relationships to diagnose if there are any major model OH biases in the current version. Figure S8 shows that the base model can capture the observed toluene:benzene relationship, in terms of both emission ratios and their relative decay rates. This agreement indicates the reproduction of OH level in the model and future analysis is needed for evaluating current $k_{OH}$ in the model.




Further, we find that $k_{OH}$ for xylenes in recent GEOS-Chem versions has been updated based on new assumptions. The GEOS-Chem version 12.5.0 used in this analysis assigns $23.1 \times 10^{-12}$ cm$^3$ molecule$^{-1}$ s$^{-1}$ as $k_{OH}$ for xylenes, based on the assumption that $m$-xylene is the dominant isomer (Fischer et al., 2014). Other studies using the fractions of xylene isomers observed in urban atmospheres for a weighted $k_{OH}$ suggested values of $13.2–17.0 \times 10^{-12}$ cm$^3$ molecule$^{-1}$ s$^{-1}$ , about 25–40 % lower than

used here, which, if updated, would result in higher simulated xylenes (Atkinson and Arey, 2003; Hu et al., 2015a; Bates et al., 2021). Therefore, correcting $k_{OH}$ could partly reconcile the model negative bias for xylenes ER (*i.e.*, 0.15 in corrected simulations vs 0.32 ppb ppm$^{-1}$ in observations). The isomer fractional information for other lumped species and their chemistry in various environments is less known; thus future speciated measurements could help refine and assess the chemical impact of these lumped species.

**7 Model evaluation with ground-based observations**

The national wildland fire burned area in 2019 was only about half that in 2018 ([https://www.nifc.gov/fire-information/statistics/wildfires](https://www.nifc.gov/fire-information/statistics/wildfires)). This is also reflected in the different acetonitrile distributions measured between the two aircraft campaigns (median 295 ppt during WE-CAN versus 205 ppt during FIREX-AQ; Fig. S2). To examine the year-to-year variability and regional representativeness of findings inferred from the WE-CAN C-130 measurements, we expand the

analysis to observations from nine ground-based sites in 2018 and the FIREX-AQ DC-8 aircraft in 2019. The ground stations span several urban areas that are regularly affected by wildfire smoke. More importantly, the longer-term stationary measurements can provide a counter test to the contribution of the other factors from fire sub-grid features to model bias relative to aircraft observations that target the plumes (Sect. 4).

Figure 8 shows that most of the nine ground sites were heavily impacted by wildfire smoke in the 2018 summer, as indicated by elevated CO mixing ratios lasting over a few days at times to 250 ppb or higher, while the general urban background CO is about 150–200 ppb (Pfister et al., 2011; Kim et al., 2013; Lopez-Coto et al., 2020; Gonzalez et al., 2021). Using the noBB and the base simulations, we define "BB-impacted days" as days when the modeled CO daily mean is increased by more than 20 % relative to the noBB run, and the rest of the days are termed low/no smoke days. By this definition, Seattle and Denver were

least affected by BB in 2018 among the nine sites, but still experienced 7–8 BB-impacted days out of 55 days. The rest of the sites all experienced ≥ 25 BB-impacted days, according to GEOS-Chem + GFAS. In general, the base model captures the daily variation of the observed CO (R > 0.40 at all sites, with six sites having $r$ > 0.65). In Seattle and Denver, anthropogenic emissions dominate local CO abundance and variability in 2018. The US EPA NEI emission inventory appears to have spatial biases as the base simulation captures observed CO in Denver but overpredicts CO in Seattle.




Tables S5–S7 summarize the mean bias, root mean square error (RMSE), and observation:model correlations for the entire data period, BB-impacted days, and low/no smoke days. Results show that the GEOS-Chem + GFAS underpredicts observed CO at the other seven sites by 95–140 ppb for the entire period. The model negative mean biases are larger on BB-impacted days, pointing to model errors in BB related processes. The base model does overpredict a few BB-impacted events, *i.e.,* 4th

and 17th August in California (Chico, Stockton, or Fresno), likely because local meteorological processes affecting smoke transport or the timing of BB emissions of certain individual fires are not captured in the model (O'Neill and Raffuse, 2021). Even so, the simulated CO abundance is underpredicted by > 100 ppb on 40–60 % BB-impacted days for all seven sites while the model background bias (loosely calculated by 5th percentile CO mixing ratio) tends to be less than 70 ppb. Thus, similar to the findings in Sect. 4, correcting the model background CO bias (due to anthropogenic emissions or global background) is

not enough to reconcile the large model-observation discrepancy. We find that the $3 \times$ GFAS simulation systematically improves the model mean bias to various degrees across the western US for the seven fire-influenced sites without degraded correlation coefficients with observations.

## 8 Model evaluation with FIREX-AQ aircraft observations

Figures 9 and S9 show the model evaluation with FIREX-AQ DC-8 VOC observations for the western US. Observed VOC

mixing ratios during FIREX-AQ are lower than in WE-CAN for this region partly due to less BB emissions in 2019 (Sect. 3). Overall, our findings in 2019 FIREX-AQ are consistent with the 2018 WE-CAN evaluation: the base simulation tends to underestimate all observed VOCs but MEK by a factor of 2–12 in the middle to lower troposphere. When we restrict the analysis to the low/no smoke environment, the base model also underestimates OVOCs and these model negative biases tend to be 40–100 % in the entire campaign average (Fig. S10). The model improvement for primary VOCs from tripling BB

emissions is significant across the troposphere, but not as obvious as during WE-CAN due to a smaller BB emission in 2019 (Sect. 3). Both WE-CAN and FIREX-AQ observations imply that the model misses substantial sources for OVOCs particularly formaldehyde, formic acid, acetic acid, and lumped $\geq C_3$ aldehydes.

We do not attempt to evaluate the modeled ERs for FIREX-AQ because the inventories do not update ERs for different years.

Figure 7 shows the observed ERs in WE-CAN and FIREX-AQ are consistent within the combined instrument uncertainty ($\pm$ 40 %) for a majority of VOCs in western fuel types, supporting recent findings by Gkatzelis et al. (submitted). Given the observational constraints in ERs and primary BB emissions, we infer that the above missing OVOC sources in the model are most likely from photochemical reactions in smoke plumes (especially for formaldehyde, formic acid, acetic acid, and lumped $\geq C_3$ aldehydes).



## 9 Implications for total biomass burning VOC emissions in the western US

We infer the systematic underestimation of simulated CO and individual VOCs in the western US is mostly driven by the low bias of effective dry matter burned in fire-detected areas across three global BB emission inventories. This finding is also supported by the low bias of inventories' DM burned per area (Sect. 3), the analysis of fire detections, injection heights (Sect. 5), ERs from airborne measurements (Sect. 6), and additional model evaluations with long-term stationary ground measurements (Sect. 7) and aircraft observations in a different year (Sect. 8). Nevertheless, the 3 times underestimation of effective dry matter burned can be recognized as the upper limit as the model negative bias could also be attributed to the Eulerian models not being able to resolve sub-grid features such as fire plumes (Sects. 2.3 and 4). It is impossible to rule out and quantify these sub-grid uncertainties in the $0.25° \times 0.3125°$ GEOS-Chem nested simulation (Rastigejev et al., 2010; Eastham and Jacob, 2017), though our evaluation using ground measurements help increase the argument of the dry matter burned underestimation. Novel methods such adaptive grids or embedded Lagrangian plumes are needed to fully resolve local conditions of the plume in future studies.

Sensitivity tests with tripled BB emissions result in better agreement between observations and model outputs, particularly for primary VOCs. Thus, our best estimate of the BB primary emissions of the 14 modeled VOCs for the western US 2018–2019 fire seasons is 120–1020 GgC, which is 3 times the default emission estimates in 3 BB inventories. This is also ~5–30 % of the total VOC emissions in the model, based on the 2018 WE-CAN and 2019 FIREX-AQ observational constraints. However, the model still underpredicts OVOCs, even with tripled BB primary emission; we are thus unable to constrain secondary production of BB VOCs in this work.

The above BB emission estimates are derived from 14 modeled VOCs with BB representation in 3 BB inventories (Table 1). However, the total ER of these 14 BB VOCs only accounts for half of the total measured VOC ERs from 161 species observed during WE-CAN (75 ppb ppm$^{-1}$ versus 150 ppb ppm$^{-1}$; Permar et al., 2021). The uncharacterized BB VOCs in the model mean that there is a significant amount of missing reactive organic carbon fluxes in many major BB emission inventories and CTMs. Their chemical and health impacts on the regional and global scale remain largely unexplored (Permar et al., sumbitted; Carter et al., 2022). Considering both underpredicted dry matter burned and uncharacterized VOCs, we infer that BB contributed ~10–45 % (or 240–2040 GgC) of the total VOC primary emissions in the western US during 2018–2019 fire seasons, far more significant than common model representation as in Fig. 1.

## 10 Conclusions

We performed nested GEOS-Chem simulations and compared them with observations from two recent airborne campaigns and nine surface sites to constrain the BB CO and VOC emissions in the western US. We evaluated three widely used global BB emission inventories including potentially significant errors in their dry matter burned, fire detection efficiency, injection



heights, and emission ratios. Based on the model:observation comparison, we provided an updated emission estimate of BB VOCs for both modeled and uncharacterized VOCs during two different fire seasons in the western US.

In the standard GEOS-Chem, BB VOC emissions in the western US rank as third of the total VOC primary sources (including

biogenic and anthropogenic emissions). Despite large interannual variability, the western US accounted for 60–90 % of BB VOC emissions over the CONUS in 2018 and 2019. Across three global BB inventories, total BB VOC emission estimates in the western US agreed with each other within 30–40 %. However, estimates for individual VOCs can differ by up to a factor of 5 (*i.e.,* lumped $\geq C_4$ alkanes), mostly driven by regionally averaged emission ratios rather than effective biomass burned. We found that global inventories' effective DM burned per area were underestimated by a factor of 2–5 compared to previous

field measurements and model estimates in the western US.

We found that simulations driven by three different BB inventories produce similar CO and VOC abundances. The model reproduced the plume enhancements in the locations observed in WE-CAN, but showed negative biases for CO and VOCs (except MEK). Better model performance was found in relatively clean environments. By comparing BB-impacted source

enhancements between no/low smoke times and the entire campaign, we found that the model, regardless of which BB inventory was used, underestimated the BB emissions for primary compounds by a factor of 3–7; these include CO, benzene, toluene, and propane. For OVOCs that have both primary and secondary sources including formaldehyde, formic acid, acetic acid, and lumped $\geq C_3$ aldehydes, the model suggested a less important role of their direct/primary BB emissions; model:observation comparison pointed a large amount of missing secondary production in BB impacted conditions in GEOS-

Chem. Unlike other VOCs, MEK was overestimated by a factor of 2–4 throughout the middle to lower troposphere, due to the overestimation of MEK itself or its precursors in the EPA NEI and MEGAN emission inventories. Tripling the BB emissions in GFAS reproduced observed mixing ratios for primary compounds, but showed no or less significant improvement for OVOCs.

We found that the fire detection products in all the inventories detected the large fires sampled in the WE-CAN campaign. GEOS-Chem vertical profiles were not strongly sensitive to the various tested BB injection height schemes, as constrained by the observed VOC vertical profiles during WE-CAN. This is likely because strong and efficient vertical mixing during hot and dry summers in the western US dominates the vertical transport processes. However, different injection height assumptions influenced the modeled downwind surface ozone mixing ratios (*i.e.,* daily mean ozone differed by up to 14 % or 4 ppb); thus,

the influence of injection heights on surface air quality requires further investigations.

We evaluated modeled ERs with WE-CAN (and FIREX-AQ) observations and found that GFAS performs slightly better than the QFED or GFED4 inventories for both VOC-CO correlations and ER values. The GEOS-Chem + GFAS captured the observed ERs in aircraft emission transects within 30 % for formaldehyde, acetaldehyde, benzene, toluene, and lumped $\geq C_4$

alkanes, and within 50–70 % for ethane, propane, and acetone. We also found the modeled abundance and ERs of xylenes and





lumped ≥ $C_3$ alkenes are significantly underestimated across all inventory experiments, likely reflecting the uncertainty of OH reaction rate constant $k_{OH}$ used in the model.

Given that the errors in fire detection, plume injection, and ERs are relatively small, we infer that the underestimation of BB
emissions in these inventories (a factor of 3–7) is likely due to underpredicted dry matter burned. However, we cannot rule out the uncertainties of the nested GEOS-Chem (0.25° × 0.3125 °) not being able to fully resolve the sub-grid features of BB emissions. The above findings revealed by 2018 WE-CAN observational constraints are further tested for their regional representativeness and interannual variability with observations from nine western US ground sites and the 2019 FIREX-AQ airborne campaign. Compared to the ground-based "downwind" CO measurements, the GEOS-Chem + GFAS captures the
observed BB smoke events but underpredicts the mixing ratios in most cases. Tripling the BB emission reduces the model negative bias across the western US without degrading the correlation coefficients with observations. Repeating the analyses with FIREX-AQ observations also confirm the above conclusions.

Constrained by 2018 and 2019 airborne- and ground- measurements, the 14 BB VOCs included in the model contributed to
120–1020 GgC of primary emissions in the western US 2018-2019 fire seasons. However, the total emission ratio to CO of these 14 VOCs in GEOS-Chem only accounted for half of that from the 161 measured VOCs in wildfire smoke, pointing to a significant amount of uncharacterized reactive organic carbon fluxes that were missing in many current BB emission inventories and CTMs. Thus, accounting for both these missing species and underestimated DM burned, the total BB VOC emission estimates can reach 240–2040 GgC or 10–45 % of the total primary VOC emissions in the western US fire seasons,
highlighting a significant role of wildfires in the US air quality.

**Code availability**

GEOS-Chem input and code are available at https://github.com/geoschem. Corresponding analysis codes are available on request.

**Data availability**

WE-CAN and FIREX-AQ are available in the NASA LaRC Airborne Science Data for Atmospheric Composition data archive (https://www-air.larc.nasa.gov/cgi-bin/ArcView/firexaq?MERGE=1).
MBO data are available from the University of Washington research works archive: (https://digital.lib.washington.edu/researchworks/discover?scope=%2F&query=%22mt.+bachelor+observatory%22&submit=&filtertype_0=title&filter_relational_operator_0=contains&filter_0=data)





**Author contribution**

WP, VS, DK, RJY, RSH, ECA, ITK, JLC, APS, DAJ, AF, MMC, GIG, CW, and EFV measured CO and VOCs data in fields and provided input on the manuscript. LJ performed the modeling with plume injection inputs from JRP. LH and LJ formulated the research question and prepared the manuscript with contributions from all co-authors.

**Competing interests**

The contact author has declared that none of the authors has any competing interests.

**Acknowledgements**

This study was supported by NASA (# 80NSSC20M0166), NSF (EPSCoR Research Infrastructure # 1929210, AGS # 2144896, and AGS # 1950327), and Montana NASA EPSCoR Research Initiation Funding. The 2018 WE-CAN field campaign was supported by the U.S. National Science Foundation through grants AGS #1650275 (U of Montana), # 1650786
(Colorado State U), # 1650288 (U of Colorado at Boulder), # 1650493 (U of Wyoming), # 1652688 (U of Washington), # 1748266 (U of Montana), and the National Oceanic and Atmospheric Administration (Award # NA17OAR4310010, Colorado State U). The Mt. Bachelor Observatory is supported by the National Science Foundation (grant #AGS-1447832) and the National Oceanic and Atmospheric Administration (contract #RA-133R-16-SE-0758).

The authors acknowledge high-performance computing resources and support from Cheyenne (doi:10.5065/D6RX99HX) provided by the NCAR Computational and Information Systems Laboratory, sponsored by the NSF, and the U of Montana's Griz Shared Computing Cluster (GSCC). We also thank Joel A. Thornton, Teresa L. Campos, Glenn S. Diskin, Dirk Richter, Patrick R. Veres, Joshua P. Schwarz, and Donald R. Blake for providing other WE-CAN and FIREX-AQ measurements.

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

Table 1. VOC representation in the base-case GEOS-Chem simulation and WE-CAN measurements used in model evaluations

| Formula | GEOS-Chem Species | Full Name | Biomass burning (GFAS) | Biogenic (MEGAN) | Anthropogenic (NEI2011) | Instruments | Measurement uncertainty |
|---|---|---|---|---|---|---|---|
| $C_2H_6$ | C2H6 | Ethane | X | N/A | X | AWAS | 10 % |
| $C_3H_8$ | C3H8 | Propane | X | N/A | X | **TOGA** | 10 % |
| - | ALK4[a] | Lumped ≥ $C_4$ alkanes | X | X | X | AWAS/TOGA | 10 % |
| - | PRPE[a] | Lumped ≥ $C_3$ alkenes | X | X | X | AWAS | 10 % |
| $CH_2O$ | HCHO | Formaldehyde | X | X | X | **PTR**/TOGA | 40 % |
| $CH_3CHO$ | ALD2 | Acetaldehyde | X | X | X | **PTR** /TOGA | 15 % |
| - | RCHO[a] | Lumped ≥ $C_3$ aldehydes | X[b] | N/A | X | **TOGA** | 30 % |
| $C_6H_6$ | BENZ | Benzene | X | NA | X | **PTR**/TOGA | 15 % |
| $C_7H_8$ | TOLU | Toluene | X | X | X | **PTR** /TOGA | 15 % |
| $C_8H_{10}$ | XYLE[a] | Xylenes | X | N/A | X | **PTR**/TOGA[c] | 15 % |
| $C_3H_6O$ | ACET | Acetone | X | X | X | **PTR**/TOGA[c] | 15 % |
| $CH_3C(O)C_2H_5$ | MEK | Methyl Ethyl Ketone | X[b] | N/A | X | **PTR**/TOGA[c] | 15 % |



| HCOOH | HCOOH | Formic acid | X[b] | X | N/A | PTR/**I-CIMS** | 50 % |
|---|---|---|---|---|---|---|---|
| CH₃COOH | ACTA | Acetic acid | X[b] | X | N/A | **PTR** | 50 % |

Note: measurements used for figures in Sect.4 are in bold text.

[a] The speciation of lumped VOCs in observations and models are provided in Table S2.

[b] The default GFASv1.2 in the standard GEOS-Chem does not contain RCHO, MEK, HCOOH and ACTA. We incorporate their emissions by scaling CO BB emissions with corresponding ERs from Permar et al. (2021). They are 1.01 ppb ppm⁻¹ for RCHO (sum of propanal and butanal species), 0.73 ppb ppm⁻¹ for MEK, 9.5 ppb ppm⁻¹ for HCOOH, and 8.61 ppb ppm⁻¹ for ACTA.

[c] We applied 0.78/0.22, 0.65/0.35, and 0.8/0.2 ratios to the PTR-ToF-MS measurements to approximate the isomers of acetone/propanal, xylenes/ethylbenzene, and MEK/butanal. The ratios are based on the speciated isomer distribution in the smoke transects closest to the fires observed by TOGA as described by
Permar et al. (2021).

Table 2. Description of the model experiments

| Simulation name | Biomass burning inventory | Injection height scheme |
|---|---|---|
| Inventory experiments | GFAS (base) | sf2mami[a] |
| | GFED4 | surface[b] |
| | QFED | 35 % FT, 65 % PBL[c] |
| Injection experiments | GFAS | 35 % FT, 65 % PBL[c] |
| | GFAS | surface[b] |
| | GFAS | sf2mami[a] |
| | GFAS | mami[d] |
| | GFAS | apb_apt[e] |
| noBB | BB emissions turned off | N/A |
| 3 × GFAS | Tripled BB emissions | sf2mami[a] |

[a] BB emissions are evenly distributed from surface to the mean altitude of maximum injection ('mami') from GFASv1.2.

[b] BB emissions are released into the model surface layer and mixed into the atmospheric boundary layer via diffusion before advection and chemistry operators.

[c] 65 % BB emissions by mass are released within the planetary boundary layer (PBL) and 35 % are released between the top of PBL and 5500 m in the free troposphere (FT).

[d] BB emissions are released to the mean altitude of maximum injection from GFASv1.2.

[e] BB emissions are evenly distributed from the bottom to the top of the plume from GFASv1.2.




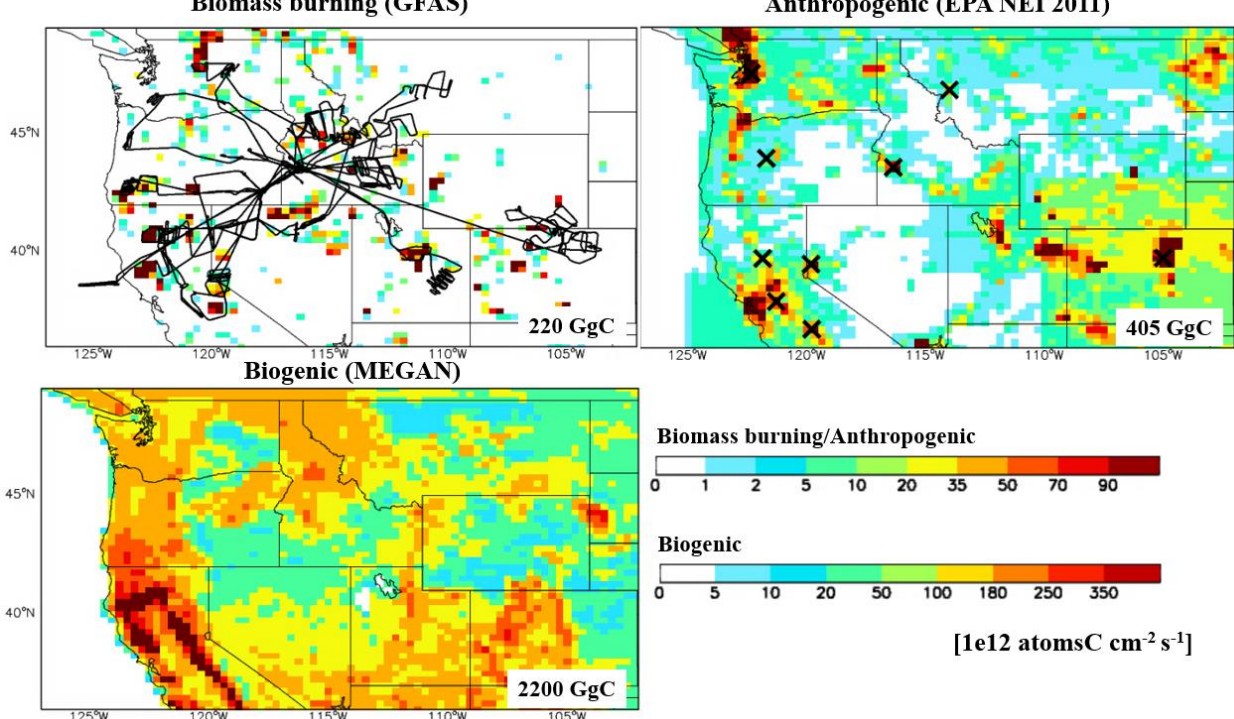

Figure 1. VOC primary emissions over the western US in the base GEOS-Chem simulation for the 2018 fire season (JJAS). Also shown are the C-130 flight tracks during WE-CAN (black lines in the top left map), and locations of the ground stations used in this study (black × symbols in the upper right map). Note the color scale for biogenic emissions (MEGANv2.1) is different from that for biomass burning (GFASv1.2) and anthropogenic emissions (US EPA NEI 2011). VOC speciation for biomass burning in the base simulation is provided in Tables 1 and S2.







Figure 2. Biomass burning VOC emission estimates for the 2018 fire season (JJAS) (black) and emission ratios (red) over the western US
in three global emission inventories. The emission ratios are regionally averaged from each inventory, and are calculated from the regression
of daily mean VOC and CO BB emission fluxes at each grid cell for the region. Error bars represent 95 % confidence intervals from the
bootstrapping resampling of the regression. We note that regionally averaged emission ratios derived from inventories might differ from
those for individual fires derived from the full chemistry simulations used in Sect. 6.



Figure 3. Median vertical profiles of CO mixing ratios in the western US during the WE-CAN aircraft campaign (July–September 2018). GEOS-Chem simulations driven by three different biomass burning emission inventories (GFED4s, GFASv1.2, and QFED2.4) are compared to observations. Also shown are two model sensitivity tests with biomass burning emission turned off (noBB) and with tripling GFASv1.2 emission (3 × GFAS). Model results are sampled along the flight tracks at the time of research flights; and observations are regridded to model resolution. Profiles are binned to the nearest 30 hPa. Horizontal bars show the 25[th]–75[th] percentile range of measurements in each vertical bin. The number of observations in each bin is given on the right side of each panel. Results are filtered to include only data where the number of datapoints for the pressure bin is larger than 10.



Figure 4. Median vertical profiles of observed VOC mixing ratios in the western US during WE-CAN. GEOS-Chem simulations driven by three different biomass burning emission inventories (GFED4s, GFASv1.2, and QFED2.4) are compared to observations. Also shown are two model sensitivity tests with biomass burning emission turned off (noBB) and with tripling GFASv1.2 biomass burning emission (3 × GFAS). Model results are sampled along the flight tracks at the time of the flights; and observations are regridded to model resolution. Profiles are binned to the nearest 30 hPa. Horizontal bars show the 25th–75th percentile range of measurements in each vertical bin. The number of observations in each bin is given on the right side of each panel. The number of observations in each bin is given on the right side of each panel. Results are filtered to include only data where the number of datapoints for the pressure bin is larger than 10.



Figure 5. Median vertical profiles of observed VOC mixing ratios in the western US for low/no smoke conditions sampled in WE-CAN. GEOS-Chem simulations driven by three different biomass burning emission inventories (GFED4s, GFASv1.2, and QFED2.4) are compared to observations. Results are filtered to include only data coincident with the bottom 25$^{th}$ percentile of observed acetonitrile, where $\Delta CH_3CN/\Delta CO$ is less than 2.01 ppb ppm$^{-1}$, and where the number of datapoints for the pressure bin is larger than 10. Model results are sampled along the flight tracks at the time of flights; and observations are regridded to model resolution. Profiles are binned to the nearest 30 hPa. Horizontal bars show the 25$^{th}$–75$^{th}$ percentile range of measurements in each vertical bin. The number of observations in each bin is given on the right side of each panel.



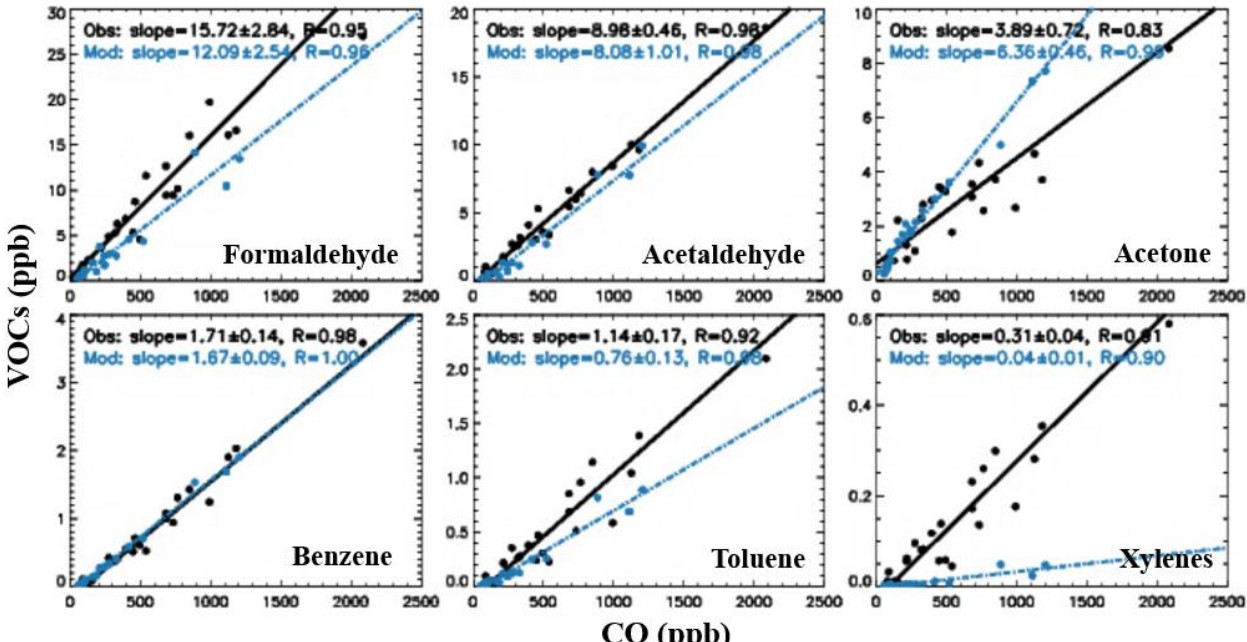

Figure 6. Biomass burning VOC emission ratios from 31 wildfire emission transects sampled on the C-130 during WE-CAN (black). Also shown are the corresponding GEOS-Chem + GFAS simulations (blue). Model results are sampled along the flight tracks at the time of flights
995 every 1 minute; and observations (and model outputs) are regridded to model resolution (5 minutes and 0.25° × 0.3125°). Lines represent the best fit of the data using the reduced major axis regression, with the regression parameters given in the equations.





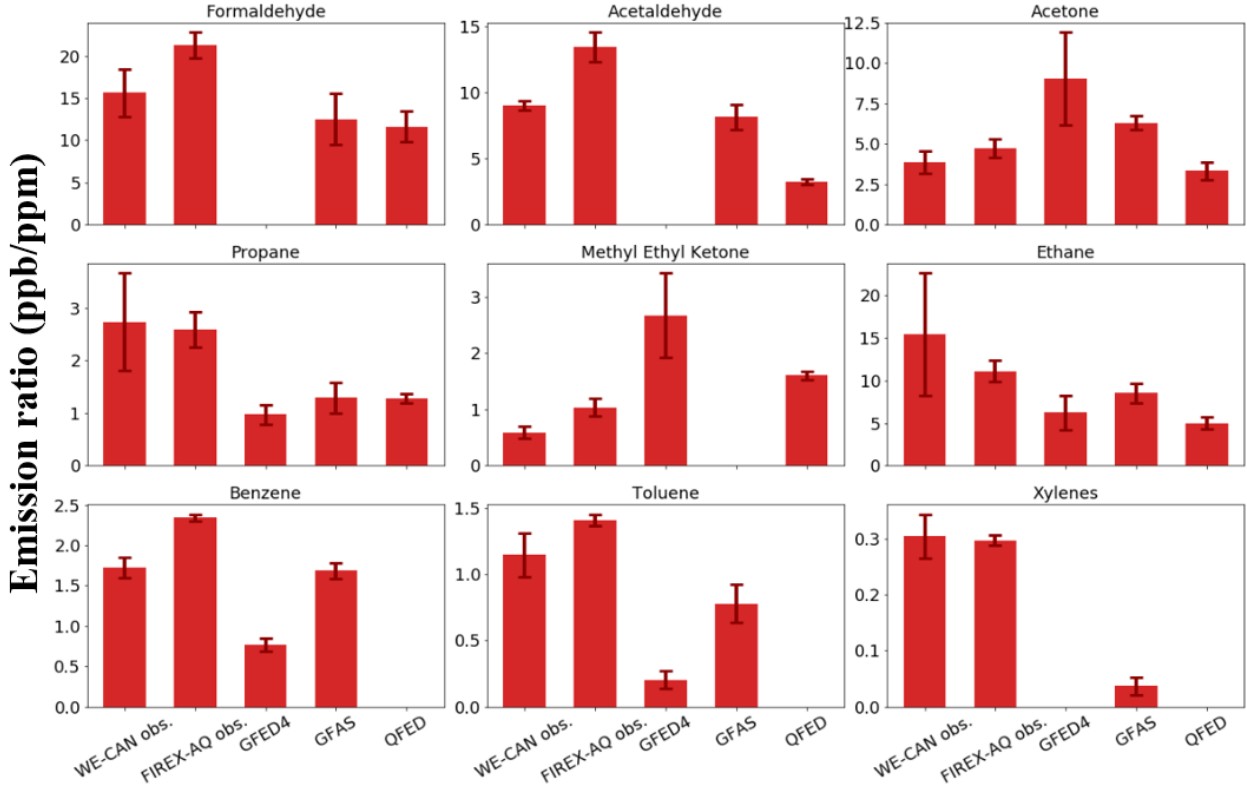

Figure 7. Summary of biomass burning VOC emission ratios for western US wildfires observed on the C-130 during WE-CAN and the DC-8 during FIREX-AQ. Also shown are the emission ratios in simulations driven by three different BB emission inventories. Values of zero indicate the species were not included in the BB emission inventory in the standard GEOS-Chem. Emission ratios are calculated from the reduced major axis regression of VOC and CO, with error bars representing the 95 % confidence interval from the bootstrapping resampling of the regression.



1005

Figure 8. Time series of daily averaged CO mixing ratios from nine ground sites in the western US during the 2018 WE-CAN campaign. Also shown are three GEOS-Chem simulations (the base simulation GFAS in blue, $3 \times$ GFAS in gray, and noBB in pink). Biomass burning emissions are injected evenly from the surface to the mean altitude of maximum injection height in the model (Table 2). The shaded area represents BB-impacted days as defined in the text. The locations of the nine ground sites are provided in Fig. 1 and Table S3.

1010



Figure 9. Median vertical profiles of observed VOC mixing ratios in the western US during the FIREX-AQ aircraft campaign (July-September 2019). GEOS-Chem driven by GFASv1.2 (base) is compared to observations. Also shown are two model sensitivity tests with biomass burning emission turned off (noBB) and with tripling GFASv1.2 emission (3 × GFAS). Model results are sampled along the flight tracks at the time of flights; and both the observations and model outputs are regridded to the model resolution. Profiles are binned to the nearest 30 hPa. Horizontal bars show the 25th-75th percentile range of measurements in each vertical bin. The number of observations in each bin is given on the right side of each panel. Results are filtered to include only data where the number of datapoints for the pressure bin is larger than 10. Observations of propane were taken from FIREX-AQ 1-minute merge data version RL, while others were from the merge data version R1.