# Peer review of "Constraining emissions of volatile organic compounds from western US wildfires with WE-CAN and FIREX-AQ airborne observations"

_EGUsphere, 2022_

## Author Comment (AC1)

We thank three reviewers for their careful consideration of our manuscript and their positive 7comments. Our point-to-point responses to individual comments are shown below in black, and reviewer comments are written in red.

Overall this is a useful addition to the literature on fire emissions of VOCs in the western US. After addressing my comments, it should be published:

Major comments:

Lines 31-33: Why is the underestimation likely due to DM underestimates and not other causes?

Biomass burning emissions are estimated as the product of the emission factor and the dry matter consumed (DM). When dividing the emission amounts by the corresponding emission factors used in inventories, we can estimate the effective DM across inventories in the region/season, even if in top-down (FRP-based) inventories DM is not explicitly calculated. Our analysis ruled out the major potential cause from emission factors/emission ratios (Section 6), since the model ERs agree with the constraints provided by WE-CAN and FIREX-AQ observations in the west. The other analyses looking at the fire detection and emission injection height (Section 5) also cannot explain the consistent negative model bias across trace gases. In addition, our calculated effective DM burned per area in the inventories indeed is too low, when compared to limited field estimates for this region (i.e., 19–28 Mg ha$^{-1}$ vs 32~44 Mg ha$^{-1}$; Section 3). All this evidence points to the DM underestimation being very likely the main reason for model low biases in trace gases.

Nevertheless, we agree that there may still be other contributions to the negative model bias. For example, smaller fires in the region that may not be captured in the global emission inventories. Additionally, Eulerian models such as GEOS-Chem are known to have trouble predicting sub-grid features like concentrated fire plumes as the fire emissions will be diluted within the coarse model grid (~25km$^2$ in this study). Therefore, we further compared the model outputs with the longer-term ground measurements across the western US, as the comparison should be less sensitive to such issues when compared to the aircraft measurements that primarily targeted fire smoke. We found that the model biases occur not only in the grids of fresh and aged plumes the aircraft sampled, but also in the widespread ground sites in the west in 2018 summer. In addition, tripling BB emissions generally improves model performance for ground sites as well. This analysis further supports that the DM is likely the dominant issue. We will further clarify our conclusions in the revision whenever possible.

206 - 224: Ideally all inventories will be used with the same vertical injection scheme no matter what the baseline in GEOS-Chem is. Can you clarify that you used the same across all three or four - either by putting all into the boundary layer or using the GFAS or QFED default schemes for all of them? Although the baseline in GEOS-Chem may use different approaches, that's just because different researchers have used them for different purposes, not because that is a scientifically appropriate approach. Furthermore, these default schemes have changed over time and used to all emit into the boundary layer with the option to turn on the Fisher et al. (2014)

scheme for all/any of them. It should not make much of a difference as you show later in the manuscript, but it would be better to ne consistent.

Also the diurnal representation should really be standardized across the inventories no matter what is in the baseline GEOS-Chem - otherwise that's another variable that's not consistent across datasets, but which could be. It would likely make sense to apply the WRAP diurnal cycle to GFED for consistency sake.

For injection height, we totally agree with the reviewer's comment that ideally all base simulations (Inventory experiments in Table 2) should use the same injection scheme. But instead, we kept the default injection scheme settings in inventory experiments, and designed another set of experiments using GFAS and varying only the injection schemes. We did not pursue the set of experiments with same injection but varying emissions as the reviewer suggested, because we did not see large impacts of these injection assumptions on the simulated vertical distribution of trace gases. As the reviewer pointed out, it should not make much of a difference. Nevertheless, with the existing simulations we have (Table 2), we are able to shed some light on and confirm this too. Figs. X1 and X2 compare GFED and GFAS when both use surface injection and then QFED and GFAS when both use PBL injection. They show that the model is indeed not sensitive to the injection schemes as examined by the campaign averaged vertical profiles at least for the western US. The comparison of simulations with the same injection but various inventories further supports the conclusion (Figs. X1 and X2). As pointed out in the manuscript, we do think the injection scheme could still affect the model performance at the surface or for long-range transport and this would require further investigation.

Regarding diurnal representation, plume-targeting aircraft campaigns like WE-CAN and FIREX-AQ are not designed to resolve the diurnal pattern of fire emissions and they only sampled several hours for each flight. That was never the intention of our analysis. Instead, we focused on the size of the fire emissions in the west, which represents the first-order question to resolve. A recent model evaluation found that the model showed no-to-little sensitivity to diurnal representation at a campaign average for both WE-CAN and FIREX-AQ, at least for the western US plumes (Tang et al., 2022). Thus, we did not expect that diurnal representation would affect the afternoon vertical profiles in this work. However, as pointed out by the reviewer and many other studies, we agree that the diurnal representation can affect the model performance for surface simulations. For complete transparency of this analysis, we added the information on the diurnal representation used in all our simulations in Table 2.

[Figure]

Figure X1. Median vertical profiles of observed VOC mixing ratios in the western US during WE-CAN. GEOS-Chem simulations driven by GFED4s and GFASv1.2 emissions, but using the same surface injection scheme are compared to observations. Model results are sampled along the flight tracks at the time of the flights; and observations are regridded to model resolution. Profiles are binned to the nearest 30 hPa. Horizontal bars show the 25[th]-75[th] percentile range of measurements in each vertical bin. The number of observations in each bin is given on the right side of each panel. Results are filtered to include only data where the number of datapoints for the pressure bin is larger than 10.

[Figure]

Figure X2. Median vertical profiles of observed VOC mixing ratios in the western US during WE-CAN. GEOS-Chem simulations driven by QFED2.4 and GFASv1.2 but using the same PBL injection scheme are compared to observations. Model results are sampled along the flight tracks at the time of the flights; and observations are regridded to model resolution. Profiles are binned to the nearest 30 hPa. Horizontal bars show the 25th-75th percentile range of measurements in each vertical bin. The number of observations in each bin is given on the right side of each panel. Results are filtered to include only data where the number of datapoints for the pressure bin is larger than 10.

Line 288-289: How does this plot show that regionally averaged ERs make up the majority of disagreement and not DM burned? The back of the envelope calculation that you discuss seems to indicate a large DM underestimate.

These are two different conclusions that do not conflict with each other. Figure 2, referred by the reviewer, compares the emission estimates among 3 inventories, while the back of the envelope calculation (plus the rest of model: observation comparisons) compares simulated trace gas concentrations driven by different BB emission inventories against observations. The difference in total fire VOC emissions across 3 inventories is smaller than the discrepancy between model and observation (i.e., < 40% vs 300-500%).

The regionally averaged ERs relative to CO are a function of assigned ERs for specific biome and vegetation classifications, both of which we cannot distinguish in the analysis of Figure 2. Regionally averaged ERs are calculated from the regression of daily mean VOC and CO BB

emission fluxes at each grid cell for the region from inventories. Figure X3a, using the same data from Figure 2, shows the regionally averaged ERs in three inventories follow the same trend of BB emission estimates for different VOCs (r = 0.8). Figure X3a further plots emissions against the regionally averaged ERs for all VOCs in inventories, and the slopes, once account for unit converts, are effectively the dry matter burned in the inventories. High correlations between them suggest that the discrepancy across inventories is due to the regionally averaged ERs, reflecting both vegetation classifications and assigned ERs.

The above calculation suggests that the effective DM burned differs by 40% across three emission inventories. Again, even though DM does not drive the discrepancy in speciated emissions across inventories, we note that it can explain largely the model's low biases when compared to field observations (also see the above response)

We further clarified this point by adding Fig. X3 into the supplement.

[Figure]

Figure X3. (a) VOC BB emission estimates and regionally averaged emission ratios relative to CO across three BB emission inventories in the western US in 2018 summer. Different symbols represent data from different inventories. (b) The scatterplots of emission estimates and regionally averaged emission ratios across three BB emission inventories using the same data from Panel a) and Figure 2.

Minor comments

Lines 57-58 - please add a citation.

The added references are provided below.

*Both EF and DM burned are subject to large uncertainties* (Akagi et al., 2011; Andreae, 2019; Carter et al., 2020)

Line 70: the comma before "even though" should be a semicolon

Edited.

Line 92: missing a semicolon between "estimates" and "thus"

Edited.

Line 99: I'm not sure that I agree with the parenthetical "(more accurate)" here. Perhaps the top-down estimates match observations better in some regions, but it may be "correct" for the wrong reasons, such as how QFED scales up emissions to match AOD, which itself may be biased high for other reasons, such as optical property uncertainties.

Thanks for pointing it out. We softened the language and deleted the parentheses in the revision.

Figure S7: I'm fairly sure that with GFED4s, GEOS-Chem does include EFs for lumped alkanes and also alkenes per the van Der Werf spreadsheet that helps us calculate them. Please confirm.

Yes, GFED4 indeed includes emission fluxes of lumped alkanes and alkenes as the model input for BB emissions. The reason for not including lumped alkanes and alkenes is that there are likely other model errors for these VOCs that skewed the correlation between these simulated VOCs and CO mixing ratios ($R^2$ below our arbitrary threshold 0.4) in these emission transects which are supposed to be mostly affected by fire smoke. See Figures X4 (and X5) for examples. For such cases, the derived slopes cannot represent the actual emission ratios in the model and thus they are not included in the analysis. Granted, should we use a more relaxed threshold or be pickier on data selection and do that on a case-by-case basis, we would be able to examine these species' ER. Take lumped alkenes as an example. Removing the 'outliers' (top left) results in an ER of 1.07 ppb/ppm for lumped alkanes (Fig. X4), which is consistent with the values in Fig. S3. However, we chose a more standardized approach and applied it to all species/simulations in this analysis.

To avoid confusion, we added this information in the captions of Figures 7 and S8, and it now reads:

*Figure 7. Summary of biomass burning VOC emission ratios for western US wildfires observed on the C-130 during WE-CAN and the DC-8 during FIREX-AQ. Also shown are the emission ratios in simulations driven by three different BB emission inventories. Model results are sampled along the flight tracks at the time of flights every 1 minute; and observations (and model outputs) are regridded to model resolution (5 minutes and 0.25° × 0.3125°). Emission ratios are calculated from the reduced major axis regression (RMA) of VOC and CO, with error bars representing the 95 % confidence interval from the bootstrapping resampling of the regression. Values of zero indicate the species were either not included in the BB emission inventory in the standard GEOS-Chem or the ER calculation fails to reach the statistical threshold ($R^2 < 0.4$) in the RMA regression.*

[Figure]

Figure X4. Modelled biomass burning VOC emission ratios from 31 wildfire emission transects on the C-130 during WE-CAN. The simulation is driven by GFED4 as the biomass burning emission input. Model results are sampled along the flight tracks at the time of flights every 1 minute; and observations (and model outputs) are regridded to model resolution (5 minutes and $0.25° \times 0.3125°$). Lines represent the best fit of the data using the reduced axis regression, with the regression parameters given in the equations.

Line 324: "being" should be "is"

Edited

Line 363: should be "Campaigns targeting plumes…" And adding "like WE-CAN" would also be helpful context here.

Edited

Line 392: please don't use "efficient" twice in one sentence.

Edited. We changed the "efficient vertical mixing" into the "strong vertical mixing".

Figure 7: I'm fairly sure that GFED has EFs for acetaldehyde and xylenes. Please check and add.

See the response to the previous question. Yes, GFED indeed has EFs for acetaldehyde and xylenes. As explained in the previous comment, other model errors likely blur the ER calculations for acetaldehyde and xylenes thus we do not attempt to constrain their ERs. For the sake of complete transparency of this analysis, we included these scatter plots as Fig. X5 here. Secondary production of acetaldehyde, the reaction rate of OH and xylenes, and anthropogenic emission biases are all potential ways to affect simulated correlations in those fire source locations.

[Figure]

Figure X5. Modelled biomass burning VOC emission ratios from 31 wildfire emission transects on the C-130 during WE-CAN. The simulation is driven by GFED4 as the biomass burning emission input. Model results are sampled along the flight tracks at the time of flights every 1 minute; Lines represent the best fit of the data using the reduced axis regression, with the regression parameters given in the equations.

REVIEWER 2

In this manuscript, the authors evaluated model predictions of volatile organic compound emissions from wildfires in the western United States. They performed GEOS-Chem simulations using different inventories and compared predicted VOC concentrations with those measured in 2 recent field campaigns, WE-CAN and FIREX-AQ. They found that the model systemically underpredicted concentrations regardless of the inventory used, and tripling one inventory can reconcile the model-measurement difference. There are more intricate differences when comparing individual VOC species, and the authors examined those in detail as well and proposed potential reasons. The manuscript is well written and easy to follow, and the results are significant and insightful. I recommend the manuscript be published. My comments are mostly suggestions and not required changes for publication.

In Line 146, the authors mentioned that xylenes in PTR could potentially be overestimated due to fragmentation in the mass spectrometer. I think this issue is very likely and should be explored further. The authors indicated that they would explore the measurement issue in Sect 4, but this issue was not specifically looked at in Sect 4. I wonder if the abundance of oxygenated aromatic compounds from BB emissions could factor into the overestimation of xylenes.

We very much appreciate the reviewer's positive feedback and recognizing the importance of our work! It is true that we did not specifically look at the fragmentation issue. We admit that the PTR issue can factor into the high bias of xylenes. Also, the campaign-average abundance of xylenes is typically lower than PTR's detection limit. Thus, we used TOGA-measured data whenever it is available in Sect. 4. We further clarified the data usage in Sect. 2.1.

In Lines 355-360, the authors investigated why oxygenated compounds like formaldehyde, acetic acid are underestimated, even more so than other VOCs. While I agree that secondary in-plume

production is very likely, I wonder if there could be other factors that cannot be ruled out. For example, these are all water soluble compounds. Could the wet deposition be overestimated in the model? Also, could there be production of these secondary species from other non-BB species? As mentioned earlier, this is more a suggestion for a deeper investigation, and I am just curious.

We agree that there may be other factors that contribute to the low bias of secondary VOCs. However, as the analyses show in the manuscript, the combination of model underestimation of BB emissions and missing secondary production is likely the main reason. A recent box modeling study also found secondary production of OVOCs is likely missing in fire smoke environments, due to the insufficient VOCs represented in the model (Wolfe et al., 2022). We further clarified it in the revisions.

We also acknowledge that we cannot fully rule out the impact of numerical diffusion in Eulerian models such as GEOS-Chem. It is also possible that errors in the wet deposition may contribute, but previous evaluations of the model do not reveal significant errors in the wet deposition processes (*i.e.*, Luo et al., 2020; Shah et al., 2020). Such errors, if they exist, would affect all water-soluble trace gases and aerosol in a somewhat consistent way. In addition, the western US in the summer is typically under very dry conditions (which is partially why wildfires could be significant), thus large-scale wet precipitation is rare in the region during the fire season.

There is also possibility that these secondary compounds are from photochemical production in non-fire conditions, which indeed may be related to the other processes such as those the reviewer pointed out. But the model low bias in no/low BB environments is far smaller than in the BB affected conditions (Figures 5 and S10). For example, we showed that modeled formaldehyde is biased low by up to 0.5 ppb above 750 hPa in the low/no smoke environments during WE-CAN (Fig. 5). Correcting such model bias is not enough to close the model: observation gap thus we conclude those biases should all due to biomass burning related processes.

I noticed that there is always a jump in concentrations at two heights (530 hPa and 650 hPa) in the measured data. For some species, the model can reproduce this trend, but not always. Does this have to do with the injection heights? I am wondering if this will help diagnose the underestimation as well.

Those were the pressure-altitudes where the airplane typically sampled the dense smoke. The model injection height is unlikely the reason why the model can reproduce the trend for some species but not others. This is because for the specific simulation, the model uses the same injection scheme for all simulated species. Figure S6 suggested that the model can get those enhancements for all species, but to a lesser extent for the short-lived compounds such as formaldehyde, acetaldehyde, and xylenes. We also found the emitting fire emissions at the surface typically show less enhancements in the 530 hPa compared to using the injection schemes with specific representations of plume heights. We commented on this detail in the revision.

Line 421-425: I do not fully understand this. How would aromatic hydrocarbon/hydrocarbon relationship help diagnose the OH issue? Is this based on looking at the ratio of two hydrocarbons with different reactivities to look at OH exposure (i.e. hydrocarbon clock)? If not, can the hydrocarbon clock be used to assess OH exposure?

Yes, the benzene/toluene diagnosis is based on the hydrocarbon photochemical clock. The photochemical and transport processes (mainly via OH oxidation in this case) controls the slope here. Thus, the reproduction of slopes suggests that the model can reproduce the OH exposure as these slopes in models can agree with 10% of observations for both WE-CAN and FIREX-AQ campaigns.

Minor comments:

Line 106: I cannot tell if it should be "though" or "through". Both could potentially make sense.

Thanks for pointing it out. We clarified it as:

*Another recent study by Bela et al. (2022) found that the daily mean emission estimates from seven existing inventories for a case study of a western US wildfire varied by a factor of 83, despite bracketing the observed BB CO fluxes.*

Line 318: typo in xylene

Edited.

Figure 1: it may be useful to indicate the ground site locations in the biomass burning panel too to give an idea of how far these ground sites are from the wildfires

Thanks for pointing it out. The spatial ground sites were provided in anthropogenic emission panel to make the BB emission map less busy.

Figure 2: for the species plotted with no bars, they are presumably not considered by each inventory? Might be helpful to indicate with n/a.

Yes. The species plotted with no bars were not considered in each inventory. We clarified that in the revised caption of Figure 2.

REVIEWER 3

This is a reviewer comment who due to technical issues was uploaded by the editor:

This is a nice paper that addresses a topic of great interest to atmospheric chemistry community. The authors use two aircraft campaigns, ground-based measurements and a nested model with three emission inventories to examine the biomass burning emissions in western US. They find a large underestimate of VOCs by current biomass burning emission inventories, which may have a large impact on ozone and aerosol air quality. The paper is well written and suited for ACP. I only have a few comments:

1. Dry Matter (DM) vs. Emission ratios. The authors argue that "the regionally averaged ERs dominate disagreement in emission estimates for most VOCs across the three inventories". Looking at Figure 2, the difference in emission ratios seem to be small for some species. On the other hand, the authors say in Line 293 "these inventories agree on the amount of effective DM burned within 40 %". It is unclear how much DM can account for the difference. It might be useful to add a plot for DM from different emission inventories.

Our results suggested that the discrepancy of emission estimates in three inventories follows the same pattern as the discrepancy of emission ratios across different trace gases (See Fig. X3). We also found that the calculated DM burned in the western US (calculated by dividing the emission amounts by emission factors) is 61±6 Tg for GFED4, 47±5 Tg for GFAS, and 67±7 Tg for QFED in 2018. Thus, these three inventories' calculated DM burned can agree within 40 %. If DM burned is the driving factor, the emissions should consistently follow the discrepancy pattern of DM burned for all trace gases, which is not the case in this analysis. Thus, we concluded that the regionally averaged ERs dominate disagreement in emission estimates for most VOCs across the three inventories. Please also see our response to Reviewer 1 for a related question.

2. Figure 4 vs. Figure 6. In Figure 4, the authors attribute the underestimate of OVOCs in their model to large secondary sources of OVOCs in biomass burning plumes that are missing in the model. However, it is shown in Figure 6 that observed OVOCs vs. CO slopes are well reproduced by their model. Are the authors assuming that the transects sampled here in Figure 6 have no secondary production of OVOCs? Some clarification is needed to reconcile Figures 4 and 6. Also it would be good to indicate the ages of those transected plumes.

Figure 6 only plots the data in near the fire sources (<2 hour physical ages), thus the regression slopes can be considered as emission ratios with minimal secondary production of OVOCs and such slightly aged ERs are thought to be more relevant to CTM applications (Lonsdale et al., 2020). Figure 4 plotted the entire campaign dataset. Following the reviewer's suggestion, we make the following editions in the main texts.

Original text: The 31 fresh BB emission transects identified in WE-CAN and the plume samples with physical age < 1 h in FIREX-AQ are used to calculate ERs.

Edited text: *In order to calculate ERs, plume samples with physical ages less than 2 hours in the WE-CAN campaign and less than 1 hour in the FIREX-AQ campaign are used, which are deemed to be relatively fresh, with minimal or no secondary production.*

3. It might be useful to point out the photochemical lifetimes of VOCs in the model. This will help to understand the impact of 3xGFAS in Figures 4 and 9.

We do not see evidence of tripling the biomass burning (BB) emission significantly affecting the VOC photochemical lifetime in the model. As the photochemical lifetime is the inverse of the product of reaction rate constant and OH concentration (tau = 1 / (k*OH), thus affecting OH concentration would affect the VOC lifetime. Based on the reviewer's suggestion, we first investigated how 3xGFAS would affect the modelled VOC lifetimes near the fire sources, as such an impact, if it exists, would be more evident in the source regions. We found that ERs between GFAS and 3xGFAS can agree within 20 %, indicating 3xGFAS doesn't change much of VOC lifetimes in the source region (Fig. X6). Additionally, we also found that both GFAS and 3xGFAS simulations can reproduce the same level of OH exposure in the WE-CAN and FIREX-AQ by reproducing the hydrocarbon: hydrocarbon relationship in the entire campaign datasets (Fig. X7).

Overall, we think that the 3xGFAS is still an overall small perturbation to the model photochemistry. Indeed, here the 3xGFAS only changed the fire VOC (and CO) emissions, which are still not the dominant sources in the western US during summer even in the active fire year.

[Figure]

Figure X6. Biomass burning VOC emission ratios from 31 wildfire emission transects sampled on the C-130 during WE-CAN (black). Also shown are the corresponding GEOS-Chem + GFAS simulations (blue) and GEOS-Chem + 3xGFAS simulations (grey). Model results are sampled along the flight tracks at the time of flights every 1 minute; and observations (and model outputs) are regridded to model resolution (5 minutes and 0.25° × 0.3125°). Lines represent the best fit of the data using the reduced major axis regression, with the regression parameters given in the equations.

[Figure]

Figure X7. Relationship between benzene and toluene in the western US during WE-CAN (left) and FIREX-AQ (right). Data are plotted on a log-log scale, with observations in black, GEOS-Chem + GFAS simulations in blue, and GEOS-Chem + 3xGFAS simulations in gray. Model results are sampled along the flight tracks at the time of flights; and both the observations and the model outputs are regridded to model resolution (5 min and 0.25° × 0.3125°). The regression parameters shown represent the best fit of the data using the reduced major axis regression, corresponding to the relationship between $\log_{10}$(benzene) and $\log_{10}$(toluene). The regression parameters are derived when both benzene and toluene are above the LoD of 30 ppt.

4. What is the difference between QFED and GFAS despite that they both use FRP? This may help the reader to better understand the paper. Why are there missing VOC species in some inventories in Figures 2 and 7?

Despite the utilization of the same FRP product from MODIS (i.e., MOD/MYD14), there are notable differences in the input data used to drive biomass burning (BB) emissions in QFED and GFAS. These differences include the utilization of varying land cover products, source of emission factor, treatment of clouds-polluted observational gaps, and the treatment of the emission coefficient relating the FRP to the amount of dry mass consumed.

For instance, QFED employs three biome groups to represent all global biomass: tropical forest, extratropical forest, and savanna/grass, while GFAS includes an additional category of peat. Also, the fuel distributions are distinct, as different products are applied. Furthermore, the emission factors (EFs) in QFED are primarily sourced from (Andreae and Merlet, 2001), whereas GFAS has incorporated updates from literature through 2009. Besides, different assumptions are made to fill the observation gaps created by clouds. QFED employs a sequential approach to tune the gaps based on MODIS aerosol optical thickness (AOT) and GFAS utilizes a Kalman filter and a system model that assumes persistence of FRP. Lastly, QFED and GFAS employ different methods to estimate the emission coefficient. QFED derives the coefficient by comparison of MODIS and GEOS aerosol optical depth (AOD), while GFAS calculates the coefficient via regression between FRP dry matter and combustion rate of GFED v3.1 in eight biome types. All these eventually lead to the differences of emission estimates between QFED and GFAS.

We have briefly included related information whenever possible to provide more context to help readers to understand Figures 2 and 7.

Regarding why there are missing VOC species in some inventories, please see the response to Reviewer 2. In short, these VOC species are either not included in certain inventories (Fig. 2), or are excluded from the analysis due to their poor correlations with CO in the model that don't meet our statistical threshold (Fig. 7). To make it clearer to readers, we followed the reviewer's suggestion and edited the corresponding captions of Figs. 2 and 7. Similar edits are also applied to supplement figures.

**References mentioned in the response:**

Akagi, S. K., Yokelson, R. J., Wiedinmyer, C., Alvarado, M. J., Reid, J. S., Karl, T., Crounse, J. D., and Wennberg, P. O.: Emission factors for open and domestic biomass burning for use in atmospheric models, Atmos. Chem. Phys., 11, 4039–4072, https://doi.org/10.5194/acp-11-4039-2011, 2011.

Andreae, M. O.: Emission of trace gases and aerosols from biomass burning - An updated assessment, Atmos. Chem. Phys., 19, 8523–8546, https://doi.org/10.5194/acp-19-8523-2019, 2019.

Andreae, M. O. and Merlet, P.: Emission of trace gases and aerosols from biomass burning, Global Biogeochem. Cycles, 15, 955–966, https://doi.org/10.1029/2000GB001382, 2001.

Carter, T. S., Heald, C. L., Jimenez, J. L., Campuzano-Jost, P., Kondo, Y., Moteki, N., Schwarz, J. P., Wiedinmyer, C., Darmenov, A. S., Da Silva, A. M., and Kaiser, J. W.: How emissions uncertainty influences the distribution and radiative impacts of smoke from fires in North America, Atmos. Chem. Phys., 20, 2073–2097, https://doi.org/10.5194/ACP-20-2073-2020, 2020.

Lonsdale, C. R., Alvarado, M. J., Hodshire, A. L., Ramnarine, E., and Pierce, J. R.: Simulating the forest fire plume dispersion, chemistry, and aerosol formation using SAM-ASP version 1.0, Geosci. Model Dev., 13, 4579–4593, https://doi.org/10.5194/GMD-13-4579-2020, 2020.

Luo, G., Yu, F., and Moch, J. M.: Further improvement of wet process treatments in GEOS-Chem v12.6.0: Impact on global distributions of aerosols and aerosol precursors, Geosci. Model Dev., 13, 2879–2903, https://doi.org/10.5194/GMD-13-2879-2020, 2020.

McGillen, M. R., Carter, W. P. L., Mellouki, A., J. Orlando, J., Picquet-Varrault, B. n. dict., and J. Wallington, T.: Database for the kinetics of the gas-phase atmospheric reactions of organic compounds, Earth Syst. Sci. Data, 12, 1203–1216, https://doi.org/10.5194/ESSD-12-1203-2020, 2020.

Shah, V., Jacob, D. J., Moch, J. M., Wang, X., and Zhai, S.: Global modeling of cloud water acidity, precipitation acidity, and acid inputs to ecosystems, Atmos. Chem. Phys., 20, 12223–12245, https://doi.org/10.5194/ACP-20-12223-2020, 2020.

Tang, W., Emmons, L. K., Buchholz, R. R., Wiedinmyer, C., Schwantes, R. H., He, C., Kumar, R., Pfister, G. G., Worden, H. M., Hornbrook, R. S., Apel, E. C., Tilmes, S., Gaubert, B., Martinez-Alonso, S.-E., Lacey, F., Holmes, C. D., Diskin, G. S., Bourgeois, I., Peischl, J., Ryerson, T. B., Hair, J. W., Weinheimer, A. J., Montzka, D. D., Tyndall, G. S., Campos, T. L., and Tang, W.: Effects of Fire Diurnal Variation and Plume Rise on U.S. Air Quality During FIREX-AQ and WE-CAN Based on the Multi-Scale Infrastructure for Chemistry and Aerosols (MUSICAv0), J. Geophys. Res. Atmos., 127, e2022JD036650, https://doi.org/10.1029/2022JD036650, 2022.

Wolfe, G. M., Hanisco, T. F., Arkinson, H. L., Blake, D. R., Wisthaler, A., Mikoviny, T., Ryerson, T. B., Pollack, I., Peischl, J., Wennberg, P. O., Crounse, J. D., St. Clair, J. M., Teng, A., Huey, L. G., Liu, X., Fried, A., Weibring, P., Richter, D., Walega, J., Hall, S. R., Ullmann, K., Jimenez, J. L., Campuzano-Jost, P., Bui, T. P., Diskin, G., Podolske, J. R., Sachse, G., and Cohen, R. C.: Photochemical evolution of the 2013 California Rim Fire: Synergistic impacts of

reactive hydrocarbons and enhanced oxidants, Atmos. Chem. Phys., 22, 4253–4275, https://doi.org/10.5194/ACP-22-4253-2022, 2022.